# Non-Redundant Essential Roles of Proteasomal Ubiquitin Receptors Rpn10 and Rpn13 in Germ Cell Formation and Fertility

**DOI:** 10.3390/cells14100696

**Published:** 2025-05-12

**Authors:** Wan-Yu Yue, Yi Zhang, Tian-Xia Jiang, Xiao-Bo Qiu

**Affiliations:** Ministry of Education Key Laboratory of Cell Proliferation & Regulation Biology, College of Life Sciences, Beijing Normal University, 19 Xinjiekouwai Avenue, Beijing 100875, China

**Keywords:** primordial germ cell, Rpn13, Rpn10, ubiquitin, proteasome, spermatogenesis, oogenesis, fertility

## Abstract

Primordial germ cells (PGCs) undergo proliferation, migration, and sexual differentiation to produce gonocytes, which eventually generate germ cells. The proteasome, which degrades most cellular proteins, is a protein complex with dozens of subunits. The proteasomal ubiquitin receptors Rpn10 and Rpn13 have been shown to play partially overlapping roles in binding ubiquitin chains in vitro and in liver function in vivo. However, the specific role of Rpn10 and Rpn13 in germ cell production remains unclear. We show here that Rpn10 and Rpn13 are each essential for germ cell production and fertility. The conditional deletion of either Rpn10 or Rpn13 in PGCs results in infertility in both male and female mice. Germ cells in testes and ovaries all decreased dramatically in the Rpn13 conditional knockout (*cKO*) mice. Specifically, the deletion of Rpn13 in PGCs disrupts the assembly of the 26S proteasome, reduces the number of PGCs, and blocks the meiosis of spermatocytes at the zygotene stage during prophase I; on the other hand, the deletion of Rpn10 in PGCs sharply reduces PGC migration. These results are important for understanding the roles of Rpn10 and Rpn13 in germ cell development and related reproductive diseases.

## 1. Introduction

Spermatogenesis and oogenesis enable mammals to produce mature sperm and eggs, which are derived from spermatogonial stem cells and primordial follicles, respectively. Both spermatogonial stem cells and primordial follicles initially arise from approximately six primordial germ cells (PGCs) that appear in the proximal posterior ectoderm on embryonic day 6.25 (E6.25) in mice [1]. Following proliferation, specialization, and migration, PGCs localize to the genital ridge and then undergo sex differentiation into gonocytes at E13.5. The female gonocytes go through the preleptotene stage, zygotene stage, and pachytene stage of meiosis and finally stay at the diplotene stage [2]. A portion of the primordial follicles resume meiosis and continue to grow until they enter puberty after birth [3]. The primordial follicles produced by the breakdown of germinal vesicle (GVBD) form primordial follicles pool for their lifelong production. Male gonocytes undergo mitotic arrest after sex differentiation and reenter mitosis until 1–2 days after birth and migrate to the lateral side of the testicular cord, where spermatogonia and spermatocytes are produced [4]. Following two rounds of meiosis, haploid round spermatids are formed, and then mature sperm with acrosome and flagella are produced after spermiogenesis [5].

Proteasomes are responsible for the degradation of most cellular proteins with four types of activators, namely the 19S regulatory particle (RP), PA28α/β, PA28γ, and PA200 [6,7]. The 26S proteasome, which degrades ubiquitinated proteins, is a ~2500 kDa protein complex that consists of the 20S core particle (CP) and the 19S regulatory particle. The CP is composed of α and β subunits, which form a cylindrical structure of heteroheptameric α_1–7_β_1–7_. In the pachytene stage of mammalian mouse testis, a large number of α4 variant α4s appear to form the testis-specific spermatoproteasome [8]. The subunits in the 19S particle control substrate recognition, unfolding, and transport into the open gate of the 20S core particle. The recruitment of the ubiquitinated substrates in the 26S proteasome is mediated directly by the proteasomal ubiquitin receptors, including Rpn1, Rpn10, and Rpn13 [6]. Each of these ubiquitin receptors seems to play distinct roles in binding ubiquitin chains. Rpn10 has two UIM domains, which bind K48 and K63 ubiquitin chains with similar affinity [9], whereas the N-terminal PRU domain of Rpn13 tends to bind the K48-linked ubiquitin chain [10]. Rpn10 and Rpn13 bind distal and proximal ubiquitin, respectively, though they can simultaneously bind the K48-linked ubiquitin chain [11]. On the other hand, Rpn1 might act as a co-receptor of Rpn10 primarily for binding the K63 ubiquitin chain [12]. Rpn10 and Rpn13 have overlapping functions, but Rpn1 is indispensable for development [13]. The conditional deletion of either Rpn10 or Rpn13 in the liver results in mild liver damage, but their simultaneous deletions lead to severe liver damage, suggesting that Rpn10 and Rpn13 play redundant roles in the liver [9]. The global deletion of Rpn13 blocks both oogenesis and spermatogenesis, leading to infertility in mice [14], whereas the global deletion of Rpn10 results in mouse embryonic lethality [15]. However, the specific roles of Rpn10 and Rpn13 in germ cell production remain unclear. We and others have previously identified Rpn13 as a subunit of the mammalian 26S proteasome [16,17,18]. This study demonstrates that the conditional deletion of Rpn10 or Rpn13 in PGCs does not block the formation of PGCs, though leading to infertility in both male and female mice. While the deletion of Rpn10 in PGCs sharply reduces PGC migration, the deletion of Rpn13 in PGCs disrupts the assembly of the 26S proteasome, reduces the number of PGCs, and blocks the meiosis of spermatocytes at the zygotene stage during prophase I.

## 2. Materials and Methods

### 2.1. Animals

The C57BL/6J mice of *Rpn13^flox/+^* were provided by Nanjing Biomedical Research Institute of Nanjing University. The C57BL/6N mice of *Rpn10^flox/+^* were provided by Biocytogen Pharmaceuticals Co., Ltd. (Beijing, China); *Blimp1-Cre* mice were kindly provided by Professor Jinsong Li. All mice were housed in specific pathogen-free-grade animal houses under alternating conditions of 12 h of light and 12 h of darkness.

*Rpn13^flox/flox^* mice were obtained by mating *Rpn13^flox/+^* and *Rpn13^flox/+^* mice. *Rpn13^flox/flox^* and *Rpn13^flox/+^Blimp1-Cre* were further mated to generate *Rpn13^flox/flox^Blimp1-Cre* (*Rpn13-cKO*), *Rpn13^flox/+^*, *Rpn13^flox/flox^*, and *Rpn13^flox/+^Blimp1-Cre* (*Rpn13-control*) mice. *Rpn10-cKO* and control mice were obtained by similar processes. For genotyping, DNA was extracted from the mouse tail or tissue and analyzed by PCR (98 °C, 2 min; 98 °C,10 s, 60 °C, 20 s, 72 °C, 30 s, 35 cycles; 72 °C, 5 min). The primers used are listed in Appendix A.

### 2.2. Tissue Collection and Immunofluorescence or Immunohistochemical Staining

Embryos or gonads were fixed in 4% paraformaldehyde at 4 °C, dehydrated, embedded in paraffin, and sectioned (5 μm). The sections were dewaxed, hydrated, and boiled in the antigen retrieval buffer (10 mM sodium citrate, pH 6.0) using a microwave oven for about 10–15 min. Then, the immunohistochemical staining was performed as per the instructions of the general SP kit SP-9000 (Beijing Zhongshan-Golden Bridge Biological Technology Co., Ltd., Beijing, China). Immunofluorescence analysis was performed following washing the sections in PBS three times and blocking with goat serum in 0.2% TritonX-100. The sections were then incubated with primary antibodies at 4 °C overnight. After washing with PBS three times, the sections were incubated with secondary antibodies for 1 h at room temperature. Next, they were mounted with DAPI (5 μg/mL) after washing with PBS three times. The images were taken under a regular microscope or a confocal microscope (Zeiss, Oberkochen, Germany, LSM 700, or LSM 880). The primary and secondary antibodies used are listed in Appendix A.

### 2.3. Immunoblotting

Testes were ground in a mortar, sonicated, and then centrifugated at 16,000× *g* for 10 min. The proteins were separated by SDS-PAGE or by Native-PAGE, followed by incubating on ice for 30 min. For SDS-PAGE, the testes were homogenized and lysed in the buffer (50 mM Tris-HCl, pH8.0, 100 mM KCl, 1 mM EDTA, 1 mM EGTA, 1% TritonX-100, 2.5 mM Na_4_P_2_O_7_, 1 mM β-Glycerophosphate, 1 mM Na_3_VO_4_, and a protease inhibitor mixture). As for native-PAGE, testes were homogenized and lysed in the buffer (50 mM Tris-HCl, PH7.5, 250 mM Sucrose, 5 mM MgCl_2_, 0.5 mM EDTA, 2 mM ATP, 1 mM DTT). After proteins were transferred to the polyvinylidene fluoride (PVDF) membrane (Millipore, Burlington, MA, USA), they were blocked with 5% non-fat milk and incubated with primary antibodies at 4 °C overnight. The secondary antibodies were horseradish peroxidase-labeled anti-rat, anti-rabbit, or anti-mouse IgG. The primary and secondary antibodies used for immunoblotting are listed in Appendix A.

### 2.4. Hematoxylin and Eosin (H & E) Staining

Testes and ovaries were fixed in 4% paraformaldehyde at 4 °C overnight, dehydrated in 50–100% gradient ethanol, transparentized in xylene twice, dipped in warm wax four times, embedded in paraffin, and sectioned (5 μm). Then, the sections were dewaxed with xylene three times until the paraffin was removed, hydrated with ethanol and H_2_O three times, and stained with hematoxylin and eosin. The nucleus was stained with hematoxylin (blue), and the cytoplasm was stained with eosin (red).

### 2.5. Proteasome Activity Assay, Spermatocyte Spread and Immunolabeling

The proteasome activity assay, spermatocyte spread, and immunolabeling were performed as described previously [19]. The proteasome activity was analyzed using the peptide substrate succinyl-Lue-Leu-Val-Tyr-7-amino-4-methylcoumarin (Suc-LLVY-AMC). The primary and secondary antibodies used for immunolabeling are listed in Appendix A.

### 2.6. TUNEL Assay

Apoptosis was detected by the TUNEL assay that was performed according to the DeadEnd Fluorometric TUNEL system (Promega, Madison, WI, USA, G3250) protocol.

### 2.7. Quantification and Statistical Analysis

All data were presented as the mean ± standard error of the mean (SEM). Unless stated otherwise, the significance level between the control and the conditional knockout groups was determined by the two-tailed unpaired *t* test, and the statistical significance was defined as *p* < 0.05 (n.s., not significant; * *p* < 0.05; ** *p* < 0.01; *** *p* < 0.001). All of the images were chosen blind and randomly and quantitated by Image J (ImageJ 1.52n).

## 3. Results

### 3.1. Conditional Deletion of Rpn13 in PGCs Reduces the PGC Number at E10.5

Germ cell formation is rigorously regulated by the ubiquitin–proteasome pathway. The global deletion of the proteasomal ubiquitin receptor Rpn13 leads to infertility in both male and female mice [14]. Primordial germ cells (PGCs) are the sources of hermaphroditic germ cells and undergo germline lineage restriction, specification, migration, and proliferation. To explore the mechanism underlying the role of Rpn13 in reproduction, we selectively knocked out the *Rpn13* gene in PGCs. Using CRISPR-Cas9-assisted homologous recombination [20], two LoxP sites were inserted, respectively, into intron 2 and intron 5 of the *Rpn13* gene to generate mice carrying the *flox* allele (*Rpn13^flox/+^*) (Figure 1A).

Although Blimp1 is a key transcription factor controlling terminal plasma cell differentiation [21], it is essential for the formation of PGCs [1]. To disrupt the *Rpn13* gene in PGCs, the *Rpn13^flox/+^* mice were further mated with *Blimp1-Cre* mice, which express the Cre recombinase in PGCs driven by a Blimp1 promoter as early as E6.25 [1]. Six genotypes of *Rpn13* alleles, including *Rpn13^+/+^*, *Blimp1-Cre*, *Rpn13^flox/+^*, *Rpn13^flox/flox^*, *Rpn13^flox/+^Blimp1-Cre*, and *Rpn13^flox/flox^Blimp1-Cre*, were generated by mating the *Rpn13^flox/flox^* with *Rpn13^flox/+^Blimp1-Cre* mice, as verified by PCR (Figure 1B). Rpn13 was conditionally deleted in PGCs in *Rpn13^flox/flox^Blimp1-Cre* mice (i.e., *Rpn13-cKO*), but not in *Rpn13^flox/+^*, *Rpn13^flox/flox^* and *Rpn13^flox/+^Blimp1-Cre* mice. Stella (also known as Dppa3 or PGC7), a gene expressed in lineage-restricted germ cells, can usually mark PGCs [22]. The total number of Stella-positive PGCs had no difference between the control and the *Rpn13-cKO* embryo at E8.5 and E9.5 but was markedly reduced at both E10.5 and E11.5 in the *Rpn13-cKO* embryo (Figure 1C). Thus, the deletion of Rpn13 reduces the number of PGCs starting from E10.5, though it does not completely deplete PGCs.

### 3.2. Conditional Deletion of Rpn13 Disrupts Germ Cell Development and Leads to Infertility in Both Males and Females

We further found that both male and female *Rpn13-cKO* mice were infertile (Appendix A). Rpn13-deficient testes were much smaller with thinner seminiferous tubules, though the sizes of their epididymis were only slightly reduced (Figure 2A,B). Hematoxylin and eosin (H & E) staining showed that the seminiferous tubules in *Rpn13-cKO* testes did not contain any mature sperm, which usually possesses a flagellum- and a falciform-shaped head with the nucleus and the acrosome [23,24] (Figure 2C and Appendix A). Immunostaining with the germ cell marker antibody TRA98 [25] showed that the number of germ cells in seminiferous tubules reduced dramatically (Figure 2D,E and Appendix A). There were no detectable germ cells on most seminiferous tubule sections, though a few of them were present on certain sections. The germ cells in testes and ovaries at postnatal day 1 (P1), as marked by the TRA98 and Vasa homolog (MVH), respectively, all decreased dramatically in *Rpn13-cKO* mice (Figure 2E and Appendix A). MVH usually begins to be expressed in oocytes after PGCs colonize at the gonads and is detectable throughout the entire life of oocytes after E12.5 [26]. We found that the ovary of the *Rpn13-cKO* mice at 5 weeks was smaller than that of the control group (Figure 2G and Appendix A) with fewer MVH-positive follicles and oocytes (Figure 2H,I and Appendix A) in the *Rpn13-cKO* mice. These results demonstrate that the deletion of Rpn13 in PGCs disrupts germ cell development, leading to infertility in both male and female mice.

### 3.3. Deletion of Rpn13 Blocks Meiosis of Spermatocytes at Zygotene Stage During Prophase I

During spermatogenesis, spermatogonia undergo mitosis and differentiation into primary spermatocytes, which further undergo meiosis, leading to the formation of haploid spermatids and sperm [27]. The number of undifferentiated spermatogonia marked by the promyelocytic leukemia zinc-finger protein (PLZF) [28] had no difference between the control and *Rpn13-cKO* groups (Figure 3A). The synaptonemal complex protein 3 (SYCP3) is partially retained along chromosome arms until metaphase I, while SYCP1 is only just present in autosomes [29]. The number of SYCP3-positive cells decreased sharply (Figure 3B), and spermatocytes marked with either SYCP3 or SYCP1 were only detectable until the zygotene stage, but not the pachytene stage, during prophase I in *Rpn13-cKO* testes (Figure 3C and Appendix A). The proportions of preleptotene/leptotene, zygotene/zygotene-like, and pachytene were 26.7%, 17.8%, and 55.5%, respectively, in the control group, but their proportions became 90.8%, 9.2%, and 0%, respectively, in the *Rpn13-cKO* group (Figure 3D). The expression of phosphorylated H2AX (γH2AX) is usually strong at the leptotene stage, gradually weakened at the zygotene stage, and enriched in sex-body at the pachytene stage [30]. However, no γH2AX-positive sex-body was found, while the levels of γH2AX at the zygotene stage were reduced in *Rpn13-cKO* testis (Figure 3E and Appendix A). Finally, the number of apoptotic cells detected by the TUNEL assay in the TRA98-positive cells increased markedly, suggesting that Rpn13 cKO increases the ratio of apoptosis in spermatocytes (Figure 3F). Together, these results indicate that the conditional deletion of Rpn13 in PGCs blocks meiotic progression at the zygotene stage of prophase I in spermatocytes.

### 3.4. Conditional Deletion of Rpn13 Markedly Reduces Activity of the 26S Proteasome

The deletion of Rpn13 in PGCs dramatically reduced the levels of Rpn13 in testes (Figure 4A), apparently due to the depletion of Rpn13 in germ cells, which are the primary group of cells in the testis. Using the specific proteasomal fluorogenic peptide substrate, Suc-LLVY- AMC, we found that the activities of the 26S proteasome were sharply reduced in the *Rpn13-cKO* testes (Figure 4B,C). The deletion of Rpn13 in PGCs dramatically reduced the levels of 19S subunits, including the ubiquitin receptors (Rpn13, Rpn1, and Rpn10) and deubiquitinating enzymes (UCH37, Usp14, and Rpn11), in addition to the proteasome activator PA200, and increased the levels of 20S subunits, including β1, β7 and the spermatoproteasome subunit α4s, as analyzed by immunoblotting following native-PAGE (Figure 4D). In contrast, SDS-PAGE displays a decrease in the total level of β7, including the free form of the β7 subunit (Figure 4A). There is probably a negative feedback regulation of the free β7 expression caused by the increase in β7 in the assembled 26S proteasome. These results suggest that the conditional deletion of Rpn13 reduces proteasomal activities, probably by disrupting the assembly of the 26S proteasome.

### 3.5. Deletion of Rpn10 in PGCs Sharply Reduces PGC Migration and Leads to Infertility

Rpn10 and Rpn13 were suggested to play redundant roles in recognizing ubiquitinated proteins and maintaining cellular homeostasis [9]. The global deletion of Rpn10 leads to early-embryonic lethality in mice [15], but the specific role of Rpn10 in germ cell development remains unclear. Thus, we constructed mice with the conditional deletion of Rpn10 in PGCs. Two LoxP sites were inserted in non-coding regions between exons 1 and 6 of the *Rpn10* gene by CRISPR-Cas9. The deletion of the *Rpn10* gene in PGCs was achieved using *Blimp1-Cre* for the selective expression of Cre recombinase in PGCs (Figure 5A,B). Except for the *Rpn10-cKO* mice with the genotype of *Rpn10^flox/flox^Blimp1-Cre*, mice with other genotypes including *Rpn10^+/+^*, *Blimp1-Cre*, *Rpn10^flox/+^Blimp1-Cre*, *Rpn10^flox/+^*, and *Rpn10^flox/flox^* were also obtained. PGCs were only sporadically found at the genital ridges at E10.5 and E11.5 in *Rpn10-cKO* mice, suggesting that PGCs were mostly lost along the migration process in *Rpn10-cKO* mice (Figure 5C). Furthermore, there were few Stella-positive gonocytes in male and female gonads from the *Rpn10-cKO* mice at E13.5, as analyzed by immunostaining (Figure 5D,E). These results suggest that the specific deletion of Rpn10 in PGCs sharply reduces the migration of PGCs.

The testis of male *Rpn10-cKO* mice was apparently much smaller on postnatal day 32 (Figure 6A,B). H & E staining showed that the mouse testicular seminiferous tubules were essentially empty except for a ring of cells near the basement membrane in the *Rpn10-cKO* testis (Figure 6C). Immunofluorescence staining with the germ cell marker TRA98 revealed that there were no germ cells in the seminiferous tubules of the *Rpn10-cKO* testis (Figure 6D). Immunofluorescence staining with SYCP3 demonstrated that there were no spermatocytes in the *Rpn10-cKO* testis (Figure 6E). Thus, the conditional deletion of Rpn10 in PGCs leads to the loss of spermatocytes.

The ovaries of *Rpn10-cKO* mice were larger than those of the control mice at approximately 18 months after birth (Figure 6F), but the numbers of follicles and oocytes, which are MVH-positive, were markedly reduced in the ovaries from *Rpn10-cKO* female mice (Figure 6G and Appendix A). Consequently, there were no offspring mice following the mating of *Rpn10-cKO* female mice with wild-type male mice or the mating of wild-type female mice with *Rpn10-cKO* male mice (Figure 6H). These results suggest that Rpn10 deletion causes infertility in both male and female mice, primarily by sharply reducing PGC migration.

## 4. Discussion

The conditional deletion of Rpn10 or Rpn13 alone in the liver causes minor damage, in contrast to the severe liver damage caused by their double deletion, suggesting their redundant roles in liver development [9]. Our study shows that unlike the reduced PGC migration caused by the deletion of Rpn10 in PGCs, the deletion of Rpn13 in PGCs primarily blocks the meiosis of spermatocytes at the zygotene stage during prophase I, suggesting their non-redundant roles in germ cell formation. The role of Rpn13 in germ cells is obviously different from that in the liver, and the reason for this could be its different mechanisms in regulating the 26S proteasome. These results strongly support the notion that the roles of Rpn10 and Rpn13 vary with tissue or biological processes.

The global deletion of Rpn13 blocks both oogenesis and spermatogenesis [14], but the specific role of Rpn13 in germ cell production remains unclear. Our study demonstrates that Rpn13 is required for germ cell production in both testes and ovaries. As analyzed in the testes, the conditional deletion of Rpn13 reduces proteasomal activities, probably by disrupting the assembly of the 26S proteasome. It has been shown previously that the global deletion of Rpn10 leads to embryonic lethality in mice [15]. We show here that while the number of PGCs reduced at E10.5-11.5, very few PGCs remained at E13.5 in the mice with conditional deletion of Rpn10 in PGCs, suggesting that Rpn10 is required for the programmed migration of PGCs in the embryo.

## 5. Conclusions

The deletion of Rpn13 in PGCs disrupts the assembly of the 26S proteasome, reduces the number of PGCs, and blocks meiosis of spermatocytes at the zygotene stage during prophase I, whereas the deletion of Rpn10 in PGCs sharply reduces PGC migration. The numbers of germ cells in the testes and ovaries all decreased dramatically in Rpn13-cKO mice. The conditional deletion of either Rpn10 or Rpn13 in PGCs results in infertility in both male and female mice. In conclusion, Rpn10 and Rpn13 play non-redundant essential roles in germ cell formation and fertility in both male and female mice. These results are important for understanding the roles of Rpn10 and Rpn13 in germ cell development and related reproductive diseases.

## Figures and Tables

**Figure 1 cells-14-00696-f001:**
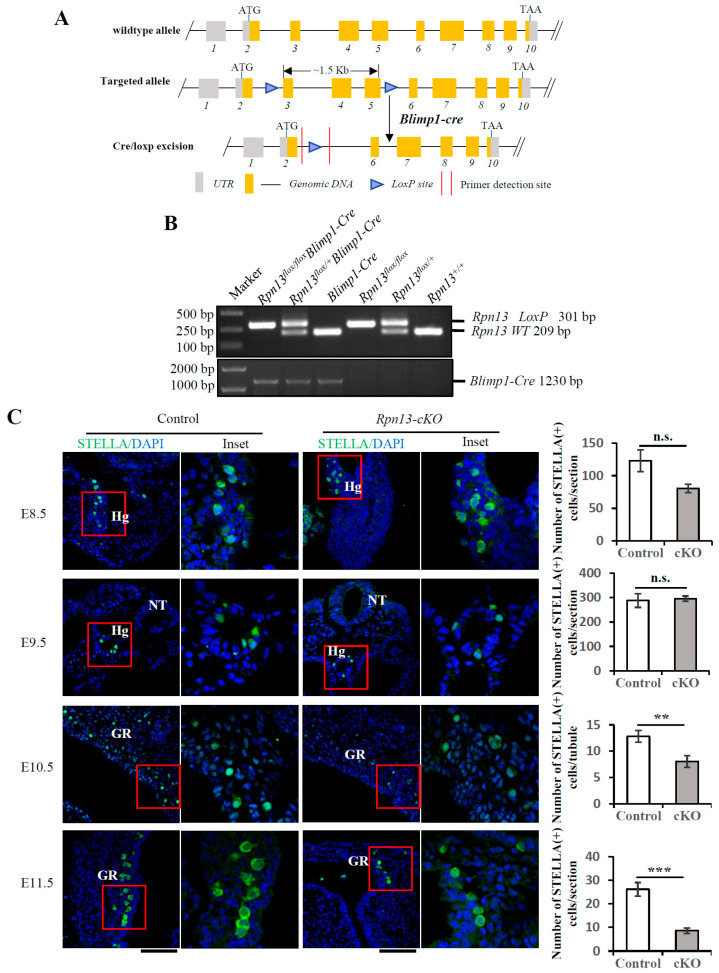
The conditional deletion of Rpn13 in PGCs reduces the PGC number at E10.5. (**A**) The strategy of Rpn13 deletion in PGC. UTR, untranslated region; ATG, translation initiation codon; TAA, translation termination codon. (**B**) Genotyping of control and *Rpn13-cKO* mice using DNA from mouse tails or tissues after birth or from the embryo. *Rpn13^flox/flox^Blimp1-Cre* are referred to as *Rpn13-cKO*. (**C**) Immunofluorescence staining for the STELLA-positive PGCs per embryo or section at E8.5, E9.5, E10.5, and E11.5 in the control (*Rpn13^flox/+^*, *Rpn13^flox/flox^* or *Rpn13^flox/+^Blimp1-Cre*) and *Rpn13-cKO* mice. DNA was stained with DAPI. The total number of PGCs at E8.5 and E9.5 were counted, while the numbers of PGCs per section were counted at E10.5 and E11.5. There were 5 and 4 embryos in the control and cKO groups at E8.5, respectively. There were 2 embryos at E9.5, 4 embryos at E10.5, and 2 embryos at E11.5 in both the control and cKO groups. Scale bar, 100 μm. Hg, hindgut; NT, neural tube; GR, genital ridge. Two-tailed unpaired *t* test, mean ± SEM. ** *p* < 0.01, *** *p* < 0.001, n.s., not significant.

**Figure 2 cells-14-00696-f002:**
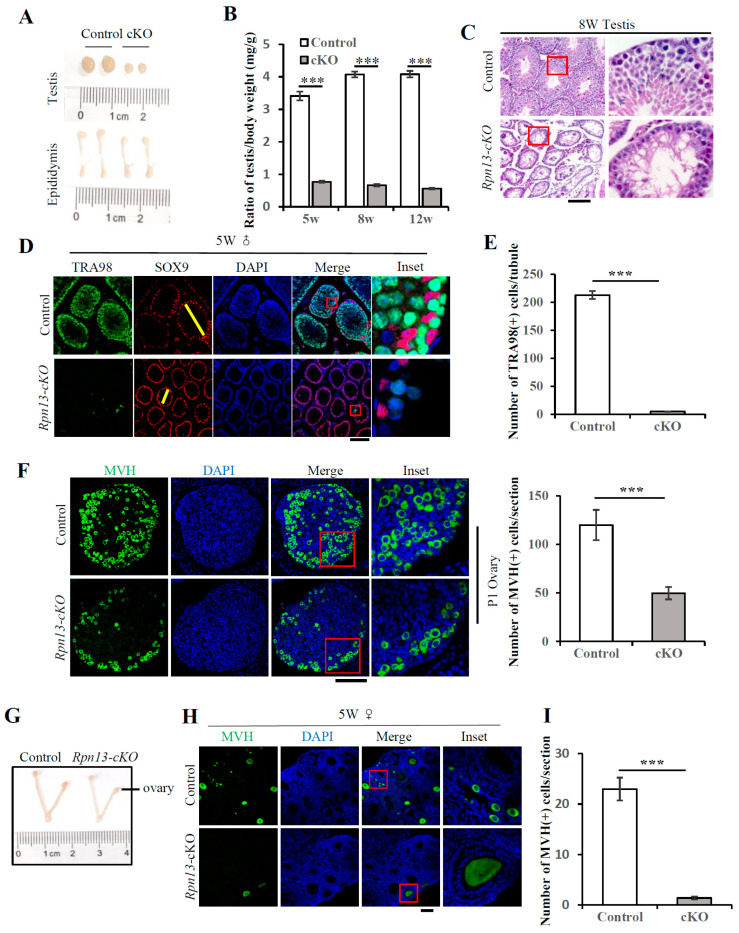
The conditional deletion of Rpn13 blocks germ cell development. (**A**) A photograph of the male testis and epididymis. (**B**) The ratio of the weight of control testes (*Rpn13^flox/+^*, *Rpn13^flox/flox^*, or *Rpn13^flox/+^Blimp1-Cre*) or *Rpn13-cKO* mice relative to the body. n = 3~5. (**C**) H & E staining of the testis sections from the control (*Rpn13^flox/+^*) and *Rpn13-cKO* mice. Scale bar, 200 μm. (**D**) The co-immunostaining of TRA98 and SOX9 in the control (*Rpn13^flox/+^*, *Rpn13^flox/flox^* or *Rpn13^flox/+^Blimp1-Cre*) and *Rpn13-cKO* testicle sections. The yellow line indicates the diameter of the seminiferous tubule. DNA was stained with DAPI. Scale bar, 100 μm. (**E**) The number of TRA98-positive cells per tubule section in the control and *Rpn13-cKO* testis. (**F**) Immunostaining of MVH on ovarian paraffin sections from control (*Rpn13^flox/flox^* or *Rpn13^flox/+^Blimp1-Cre*) and *Rpn13-cKO* mice on postnatal day 1. The numbers of MVH-positive cells per section were counted. DNA was stained with DAPI. Scale bar, 100 μm. Two-tailed unpaired *t* test, mean ± SEM. *** *p* < 0.001. (**G**) The photograph of the female control (*Rpn13^flox/+^*, *Rpn13^flox/flox^* or *Rpn13^flox/+^Blimp1-Cre*) and *Rpn13-cKO* ovaries and uteruses of 5 W old mice. (**H**) The immunostaining of MVH in the control (*Rpn13^flox/+^*, *Rpn13^flox/flox^* or *Rpn13^flox/+^Blimp1-Cre*) and *Rpn13-cKO* ovarian sections. DNA was stained with DAPI. Scale bar, 100 μm. (**I**) The number of MVH-positive cells per section in (H). Data are representative of one experiment with three independent biological replicates. Two-tailed unpaired *t* test, mean ± SEM. *** *p* < 0.001.

**Figure 3 cells-14-00696-f003:**
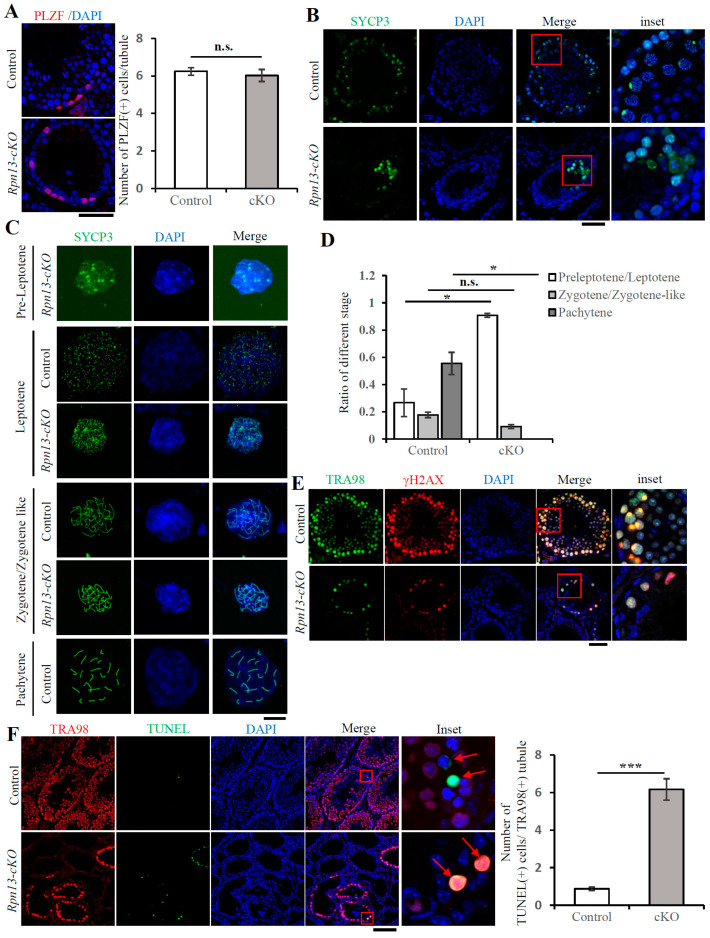
The deletion of Rpn13 blocks the meiosis of spermatocytes at the zygotene stage during prophase I. (**A**) Immunostaining of PLZF in the control (*Rpn13^flox/+^* or *Rpn13^flox/+^Blimp1-Cre*) and *Rpn13-cKO* testicle sections. The numbers of PLZF-positive cells per tubule were counted. Scale bar, 50 μm. (**B**) Immunostaining of SYCP3 in the control (*Rpn13^flox/+^* or *Rpn13^flox/flox^*) and *Rpn13-cKO* testicle paraffin sections. DNA was stained with DAPI. Scale bar, 50 μm. (**C**,**D**) The spread of chromosomes in control (*Rpn13^flox/+^Blimp1-Cre*) and *Rpn13-cKO* testes of mice at 5 W old. The ratio of chromosomes at different stages was obtained. DNA was stained with DAPI. Scale bar, 20 μm. (**E**) Co-immunostaining of TRA98 and phosphorylated H2AX in testicle paraffin sections from the control (*Rpn13^flox/+^*, *Rpn13^flox/flox^* or *Rpn13^flox/+^Blimp1-Cre*) and *Rpn13-cKO* male mice at 5 W old. DNA was stained with DAPI. Scale bar, 50 μm. (**F**) Apoptotic cells were detected by immunostaining of TRA98 and the cell apoptosis detection TUNEL assay in the control *and Rpn13-cKO* testicle frozen sections of 5W-old male mice. The red arrow indicates TUNEL-positive cells. The thickness of the section was 10 μm. The number of TUNEL-positive cells per TRA98-positive tubule in the control (*Rpn13^flox/flox^* or *Rpn13^flox/+^Blimp1-Cre*) and *Rpn13-cKO* group. DNA was stained with DAPI. Scale bar, 100 μm. Data are representative of one experiment with three independent biological replicates. Two-tailed unpaired *t* test, mean ± SEM. * *p* < 0.05, *** *p* < 0.001; n.s., not significant.

**Figure 4 cells-14-00696-f004:**
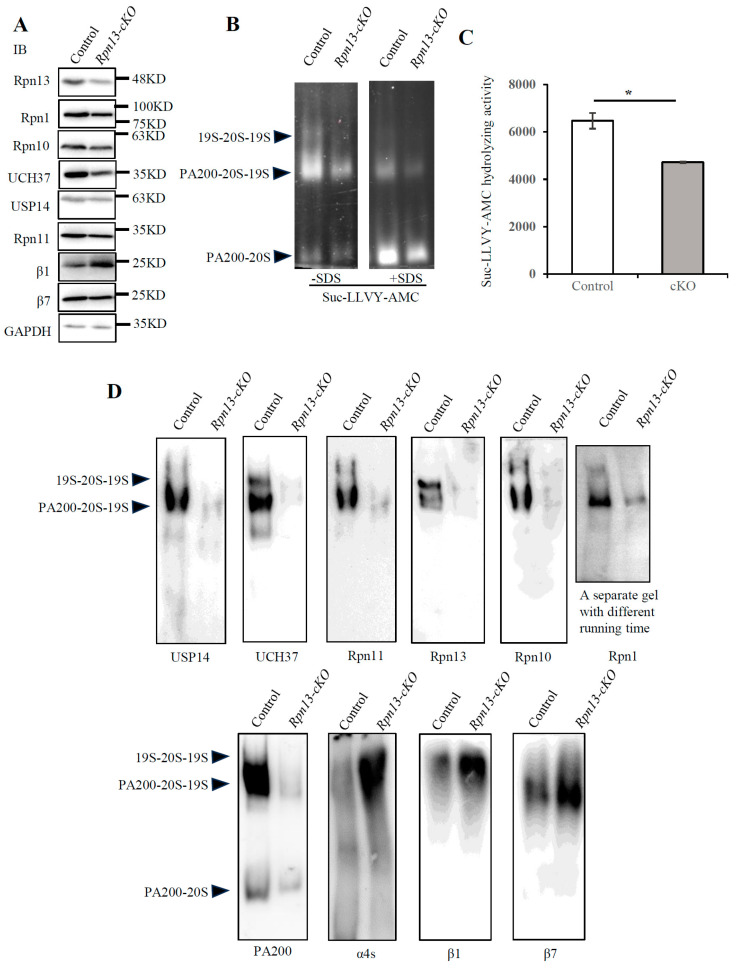
The deletion of Rpn13 in PGCs disrupts the proteasome assembly and markedly reduces the activity of the 26S proteasome. (**A**) Immunoblots were performed following the SDS-PAGE of the extracts from testes of the control (*Rpn13^flox/flox^* or *Rpn13^flox/+^Blimp1-Cre*) and *Rpn13-cKO* mice at 3 W old. (**B**) Peptidase activities were analyzed following the 4% native PAGE of the extracts from testes of the control (*Rpn13^flox/+^Blimp1-Cre*) and *Rpn13-cKO* mice at 3 W old. Proteasomal peptidase activities were detected under UV light after incubating the gel with Suc-LLVY-AMC in the absence or presence of 0.02% SDS, which stimulates 20S peptidase activities. (**C**) The 26S proteasome activity of the extracts from testes of the control (*Rpn13^flox/+^*) and *Rpn13-cKO* mice at 8 W old using the substrate Suc-LLVY-AMC. Two-tailed unpaired *t* test, mean ± SEM. * *p* < 0.05. (**D**) Immunoblotting analyses of the samples obtained were conducted, as shown in (**B**).

**Figure 5 cells-14-00696-f005:**
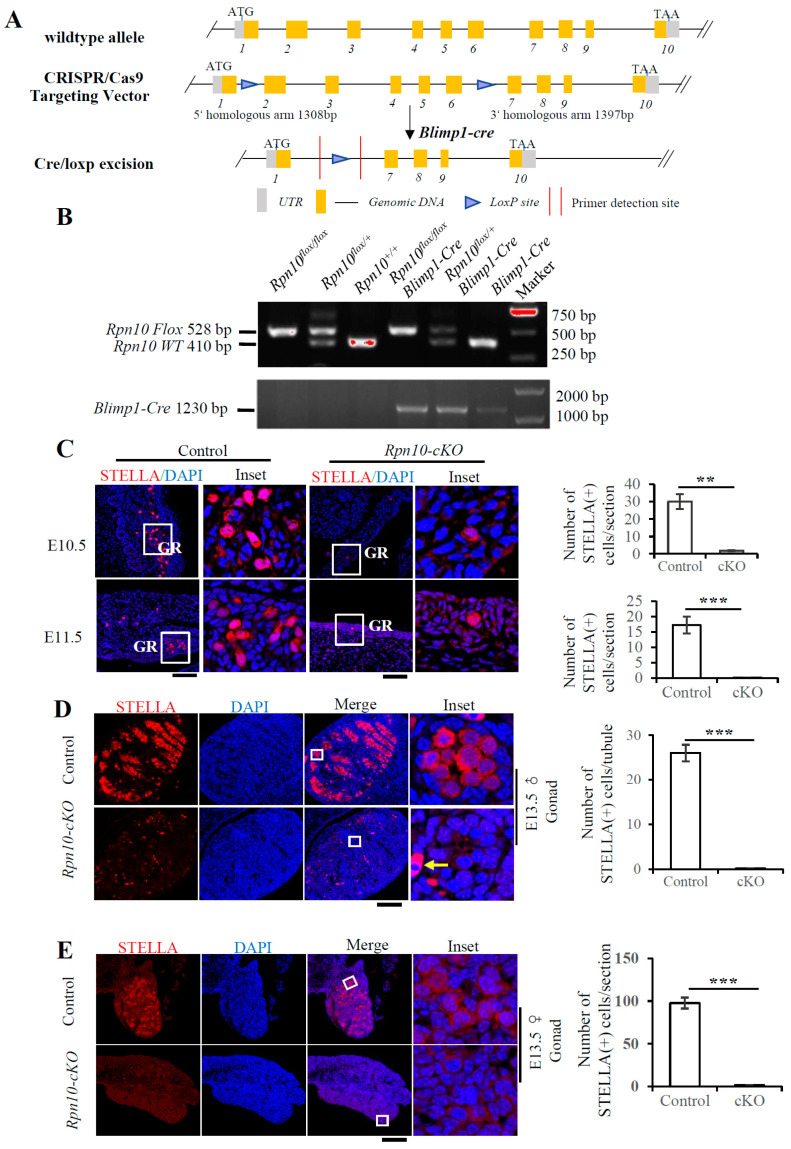
The deletion of Rpn10 in PGCs sharply reduces PGC migration. (**A**) The strategy of Rpn10 deletion in PGCs. UTR, untranslated region; ATG, translation initiation codon; TAA, translation stop codon. (**B**) The genotyping of controls and *Rpn10-cKO* mice using the DNA from the mouse tail after birth or the embryo. *Rpn10^flox/flox^Blimp1-Cre* were referred to as *Rpn10-cKO*. (**C**) Immunofluorescence staining for Stella-positive PGCs from control (*Rpn10^flox/+^* or *Rpn10^flox/flox^*) and *Rpn10-cKO* mice at E10.5 and E11.5. The number of Stella-positive cells per section was counted. DNA was stained with DAPI. Scale bar, 100 μm. Hg, hindgut; GR, genital ridge. (**D**) The immunostaining of Stella on the male gonadal paraffin sections from control (*Rpn10^flox/+^* or *Rpn10^flox/flox^*) and *Rpn10-cKO* male mice at E13.5. DNA was stained with DAPI. An arrow indicates interstitial non-specific staining. Scale bar, 100 μm. The numbers of Stella-positive cells per tubule were counted. (**E**) The immunostaining of Stella on the female gonadal paraffin sections of the control (*Rpn10^flox/+^* or *Rpn10^flox/+^Blimp1-Cre*) and *Rpn10-cKO* female mice at E13.5. There were 3 and 4 embryos in the control and cKO female groups, respectively, at E13.5. There were 2 embryos at E10.5 and E11.5 and 3 embryos at E13.5 for males in both the control and cKO groups. DNA was stained with DAPI. Scale bar, 100 μm. Two-tailed unpaired *t* test, mean ± SEM, ** *p* < 0.01, *** *p* < 0.001.

**Figure 6 cells-14-00696-f006:**
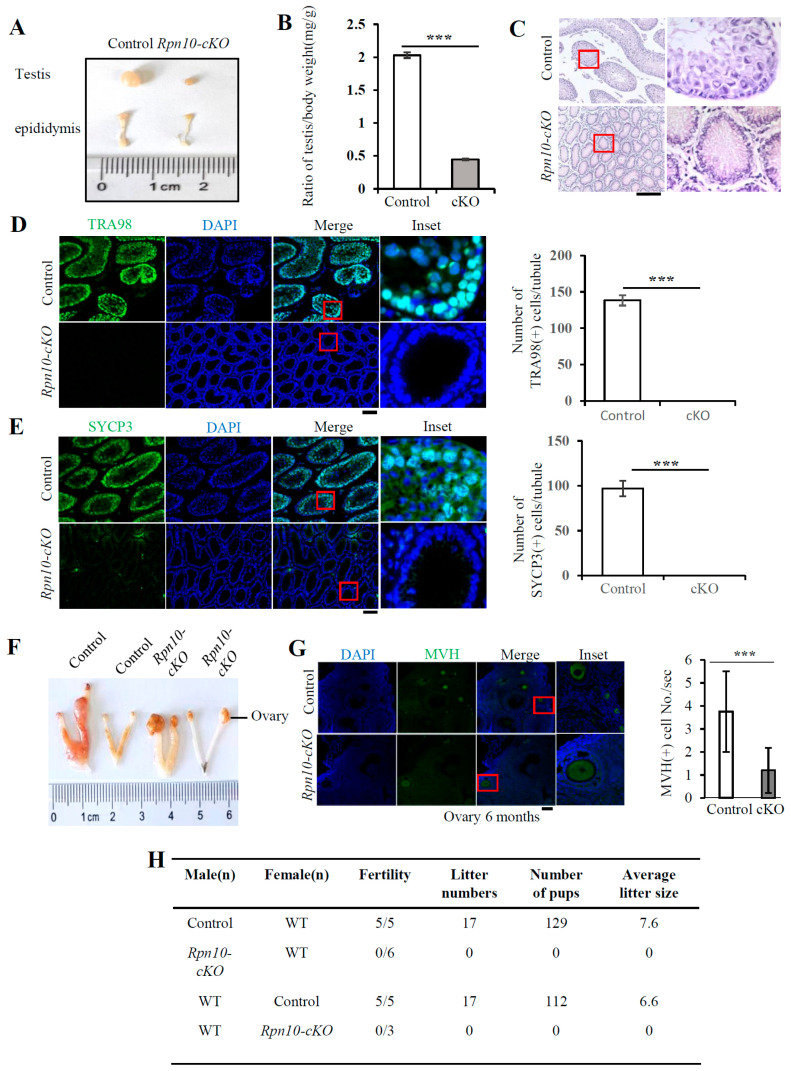
The deletion of Rpn10 in PGCs depletes germ cells and leads to infertility. (**A**) A photograph of the male testis and epididymis on postnatal day 32. (**B**) The ratio of testis weight relative to the body weight from control (*Rpn10^flox/+^* or *Rpn10^flox/flox^*) and *Rpn10-cKO* mice at 12 to 18 months old. (**C**) H & E staining of testicular paraffin sections from control (*Rpn10^flox/+^Blimp1-Cre*) and *Rpn10-cKO* mice on postnatal day 32. Scale bar, 100 μm. (**D**) Immunofluorescence staining of TRA98 from control (*Rpn10^flox/+^Blimp1-Cre*) and *Rpn10-cKO* testes on postnatal day 32. The DNA was stained with DAPI. Scale bar, 100 μm. The number of TRA98-positive cells per tubule was counted. (**E**) The immunofluorescence staining of SYCP3 from control (*Rpn10^flox/+^Blimp1-Cre*) and *Rpn10-cKO* testes on postnatal day 32. The DNA was stained with DAPI. Scale bar, 100 μm. The number of SYCP3-positive cells per tubule was counted. Two-tailed unpaired *t* test, mean ± SEM. *** *p* < 0.001. (**F**) The ovary and uterus from the control (*Rpn10^flox/+^* or *Rpn10^flox/flox^*) and *Rpn10-cKO* female mice at 12 to 18 months old. (**G**) The immunostaining of MVH-positive follicles and oocytes on ovarian paraffin sections from control (*Rpn10^flox/+^* or *Rpn10^flox/flox^*) and *Rpn10-cKO* female mice at 12 to 18 months old. The numbers of MVH-positive cells per section were counted. DNA was stained with DAPI. Scale bar, 100 μm. Two-tailed unpaired *t* test, mean ± SEM. *** *p* < 0.001. (**H**) The control and *Rpn10-cKO* mice were mated with WT mice, and the number of offspring produced was counted. The genotypes of control mice are *Rpn10*^flox/+^, *Rpn10^flox/flox^* or *Rpn10^flox/+^Blimp1-Cre*.

## Data Availability

The original contributions presented in this study are included in the article/Appendix A. Further inquiries can be directed to the corresponding author(s).

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
