# Peer review of "Non-Redundant Essential Roles of Proteasomal Ubiquitin Receptors Rpn10 and Rpn13 in Germ Cell Formation and Fertility"

_cells, 2025, doi:10.3390/cells14100696_

Round 1
Reviewer 1 Report
Comments and Suggestions for Authors
In the submitted manuscript „Non-Redundant Essential Roles Of Proteasomal Ubiquitin Receptors Rpn10 And Rpn13 In Germ Cell Formation And Fertility” the authors study the role of proteasomal ubiquitin receptors Rpn10 and Rpn13 in the formation of primordial germ cells and infertility.
It has been already shown that mice lacking Rpn13 are infertile due to defective gametogenesis (PMID: 21048919). In the submitted manuscript the authors aim to determine a specific role of Rpn10 and Rpn13 in germ cell production.
In order to achieve that, the authors generated conditional Rpn13 knockout (cKO) mice by breeding Rpn13flox/+ mice with Blimp1-Cre mice in order to remove Rpn13 in PGCs. The same strategy was applied for generating Rpn10 cKO mice.
They could show that Rpn13 cKOs exhibit reduced PGC number at E10.5 and E11.5 by quantifying a number of STELLA-positive PGC cells.
What happens at the later stages of embryonic development with PGC numbers, i.e., during mitotic PGCs (E11.5-E13.5) and mitotic arrest PGCs (E13.5E16.5)? Does PGC number ever drop to zero?
Is overall ubiquitination level increased in such Rpn13 cKO PGCs?
Do any proteins markedly accumulate in PGCs lacking Rpn13 (in comparison to wild-type PGCs) that could explain the phenotype?
In Figure 4, the authors state that “Deletion of Rpn13 in PGCs disrupts the proteasome assembly and markedly reduces the activity of the 26S proteasome.” They postulate that loss of Rpn13 “reduces proteasomal activities, probably by disrupting the assembly of the 26S proteasome.”
It was previously shown (PMID: 26222436) that the loss of Rpn13 did not affect the assembly of the 26S proteasome in liver-specific Rpn13 KO animals, with increased amount of the 26S proteasome in these samples, coupled with the increased mRNA levels of proteasome subunits, suggesting that loss of Rpn13 might induce a feedback increase in the expression of proteasome subunits. Also, yeast data show that 26S proteasome purified from strains lacking Rpn13 exhibit normal assembly of the 26S proteasome, similar to the wild-type yeast (PMID: 17499717). Furthermore, based on how Rpn13 interacts with the 19S RP, it is not very likely that Rpn13 loss affects assembly of the functional 26S proteasome. Since Rpn13 recruits UCH37 to the 19S RP, it is, however, to be expected that the amount of proteasome-bound UCH37 is decreased in Rpn13 cKO PGCs.
I would therefore suggest to provide additional experiments and/or soften that statement, both in abstract and manuscript, and to discuss it in the context of already published data about the role of Rpn13 in proteasome structure/assembly.
Is mRNA level of 19S and 20S subunits increased in Rpn13 cKO PGCs in comparison to wild-type PGCs as shown in PMID: 26222436? How does a standard SDS-PAGE/Western blot for Rpn11, USP14, UCH37 and alpha4s look like for Rpn13 cKO and wild-type PGCs (in comparison to reblotting native PAGE blots, as shown in Figure 4D)?
As for Rpn10, PGCs were only rarely found at the genital ridges at E10.5 and E11.5 Rpn10 cKO mice, with only few Stella-positive gonocytes, showing that Rpn10 loss in PGCs significantly reduces PGC migration, leading to loss of spermatocytes.
Is overall ubiquitination level increased in the surviving Rpn10 cKO PGCs?
Do any proteins markedly accumulate in PGCs lacking Rpn10 (in comparison to wild-type PGCs) that could explain the phenotype?
It would be very interesting to find out how exactly loss of Rpn13 or Rpn10 cause PGC depletion, i.e., how presumable accumulation of specific proteins, due to inefficient proteasomal degradation, leads to PGC depletion and infertility.
Minor points
Lane 49: spermatoproteasome instead of spermtoproteasome
Lane 109: Isn´t it Suc-LLVY-AMC, and not sum-LLVY-amc?
Lane 110: Word “in” is missing before TableS2.
Lanes 112-113: The sentence “The apoptosis detected by TUNEL assay was performed as the instruction in DeadEnd Fluorometric TUNEL system (Promega, G3250).” should be “The apoptosis was detected by TUNEL assay that was performed according to the DeadEnd Fluorometric TUNEL system (Promega, G3250) protocol”.
Lane 115: “All data ae presented…” should be “All data are presented…”
Author Response
Reviewer 1:
In the submitted manuscript “Non-Redundant Essential Roles Of Proteasomal Ubiquitin Receptors Rpn10 And Rpn13 In Germ Cell Formation And Fertility” the authors study the role of proteasomal ubiquitin receptors Rpn10 and Rpn13 in the formation of primordial germ cells and infertility.
It has been already shown that mice lacking Rpn13 are infertile due to defective gametogenesis (PMID: 21048919). In the submitted manuscript the authors aim to determine a specific role of Rpn10 and Rpn13 in germ cell production.
In order to achieve that, the authors generated conditional Rpn13 knockout (cKO) mice by breeding Rpn13flox/+ mice with Blimp1-Cre mice in order to remove Rpn13 in PGCs. The same strategy was applied for generating Rpn10 cKO mice.
Responses: We greatly appreciate the reviewer for critical comments and suggestions, which are addressed point-to-point as below.
They could show that Rpn13 cKOs exhibit reduced PGC number at E10.5 and E11.5 by quantifying a number of STELLA-positive PGC cells.
What happens at the later stages of embryonic development with PGC numbers, i.e., during mitotic PGCs (E11.5-E13.5) and mitotic arrest PGCs (E13.5E16.5)? Does PGC number ever drop to zero?
Responses: The PGC number in the Rpn13-cKO group reduced during mitotic PGCs (E11.5-E13.5) as represented at E12.5 and E13.5. Although we did not count the PGC numbers at E13.5 to E16.5, germ cells still existed at P1 (Figure S1B), suggesting that the PGC number did not ever drop to zero.
Is overall ubiquitination level increased in such Rpn13 cKO PGCs?
Do any proteins markedly accumulate in PGCs lacking Rpn13 (in comparison to wild-type PGCs) that could explain the phenotype?
Responses: Because the number of PGC is relatively small in embryo, we did not test the levels of overall ubiquitination or any specific proteins in Rpn13 cKO PGCs, though these comments are very important.
In Figure 4, the authors state that “Deletion of Rpn13 in PGCs disrupts the proteasome assembly and markedly reduces the activity of the 26S proteasome.” They postulate that loss of Rpn13 “reduces proteasomal activities, probably by disrupting the assembly of the 26S proteasome.”
It was previously shown (PMID: 26222436) that the loss of Rpn13 did not affect the assembly of the 26S proteasome in liver-specific Rpn13 KO animals, with increased amount of the 26S proteasome in these samples, coupled with the increased mRNA levels of proteasome subunits, suggesting that loss of Rpn13 might induce a feedback increase in the expression of proteasome subunits. Also, yeast data show that 26S proteasome purified from strains lacking Rpn13 exhibit normal assembly of the 26S proteasome, similar to the wild-type yeast (PMID: 17499717). Furthermore, based on how Rpn13 interacts with the 19S RP, it is not very likely that Rpn13 loss affects assembly of the functional 26S proteasome. Since Rpn13 recruits UCH37 to the 19S RP, it is, however, to be expected that the amount of proteasome-bound UCH37 is decreased in Rpn13 cKO PGCs.
I would therefore suggest to provide additional experiments and/or soften that statement, both in abstract and manuscript, and to discuss it in the context of already published data about the role of Rpn13 in proteasome structure/assembly.
Responses: As correctly noted by the reviewer, Rpn13 recruits UCH37 to the 19S RP in mammals. So, it is obvious that the amount of proteasome-bound UCH37 is at least decreased in Rpn13 cKO PGCs. Since yeast Rpn13 doesn’t share homology with the C-terminal region of the mammalian Rpn13, which binds UCH37, our results don’t contradict with the related results in yeast. The role of Rpn13 in germ cells is obviously different from that in liver, and the reason could be its different mechanisms in regulating the 26S proteasome. As advised by the reviewer, we have now additionally discussed this issue in the revised version (lines 342-344).
Is mRNA level of 19S and 20S subunits increased in Rpn13 cKO PGCs in comparison to wild-type PGCs as shown in PMID: 26222436? How does a standard SDS-PAGE/Western blot for Rpn11, USP14, UCH37 and alpha4s look like for Rpn13 cKO and wild-type PGCs (in comparison to reblotting native PAGE blots, as shown in Figure 4D)?
Responses: We had not measured the mRNA levels of 19S and 20S subunits in this study. Instead, we have now added SDS-PAGE/Western blot for Rpn11, USP14 and UCH37 in testes, all of which are downregulated (revised Fig. 4A).
As for Rpn10, PGCs were only rarely found at the genital ridges at E10.5 and E11.5 Rpn10 cKO mice, with only few Stella-positive gonocytes, showing that Rpn10 loss in PGCs significantly reduces PGC migration, leading to loss of spermatocytes.
Is overall ubiquitination level increased in the surviving Rpn10 cKO PGCs?
Do any proteins markedly accumulate in PGCs lacking Rpn10 (in comparison to wild-type PGCs) that could explain the phenotype?
It would be very interesting to find out how exactly loss of Rpn13 or Rpn10 cause PGC depletion, i.e., how presumable accumulation of specific proteins, due to inefficient proteasomal degradation, leads to PGC depletion and infertility.
Responses: We agree with the reviewer concerning these important issues. But we could not detect the levels of overall ubiquitination and any proteins in the Rpn10-cKO PGCs because the number of PGCs reduced dramatically in Rpn10-cKO embryo.
Minor points
Lane 49: spermatoproteasome instead of spermtoproteasome
Lane 109: Isn´t it Suc-LLVY-AMC, and not sum-LLVY-amc?
Lane 110: Word “in” is missing before TableS2.
Lanes 112-113: The sentence “The apoptosis detected by TUNEL assay was performed as the instruction in DeadEnd Fluorometric TUNEL system (Promega, G3250).” should be “The apoptosis was detected by TUNEL assay that was performed according to the DeadEnd Fluorometric TUNEL system (Promega, G3250) protocol”.
Lane 115: “All data ae presented…” should be “All data are presented…”
Responses: Thanks for these valuable suggestions, and we have now corrected all of the above issues.
Reviewer 2 Report
Comments and Suggestions for Authors
Wan-Yu Yue et al. investigated the specific roles of Rpn10 and Rpn13, which are 19S regulatory subunits of the 26S proteasome, in primordial germ cells (PGCs) using the Cre-LoxP-mediated conditional gene knockout mice. They showed that the conditional deletion of either Rpn10 or Rpn13 in PGCs results in infertility in both male and female mice and that germ cells in testes and ovaries all decreased dramatically in the Rpn13-cKO mice. Specifically, deletion of Rpn13 in PGCs disrupted the assembly of the 26S proteasome, reduced the number of PGCs, and blocked meiosis of spermatocytes at zygotene stage during prophase I, whereas deletion of Rpn10 in PGCs sharply reduced PGC migration. They stated that their findings are important for understanding the roles of Rpn10 and Rpn13 in germ cell development and the related reproductive diseases.
Major points:
Although the previous paper (Ref. 10) reported that Rpn10 and Rpn13 play a redundant role in the degradation of ubiquitinated proteins in mouse liver, Yue et al. revealed that Rpn13 and Rpn10 showed a non-redundant essential role in germ cell formation and fertility. Furthermore, the phenotypes of cKO of Rpn13 and Rpn10 appear to be a little bit different, suggesting the different roles of Rpn13 and Rpn10 in PGC. These findings are interesting and valuable for the studies on the specific roles of proteasomal Rpn13 and Rpn10 in germ cell differentiation and sexual reproduction.
Before further consideration, the authors are requested to answer the following questions.
- L232-237 and Figure 4:
It is described that “Deletion of Rpn13 in PGCs dramatically reduced the levels of 19S subunits, including the ubiquitin receptors (Rpn13, Rpn1 and Rpn10) and deubiquitinating enzymes (UCH37, Usp14 and Rpn11), in addition to the proteasome activator PA200, and increased the levels of 20S subunits, including b1, b7 and the spermatoproteasome subunit a4s, as analyzed by immunoblotting following native-PAGE (Figure 4D)”.
However, the level of b7 in Fig. 4A (immunoblotting following SDS-PAGE) appears to be slightly downregulated in Rpn13-cKO, although it appears to be upregulated in native PAGE. Why? The authors should explain this apparent discrepancy. How many proteins were loaded in each lane in SDS-PAGE and native-PAGE?
Although the results of Fig. 4A seem to be reasonable, the results (a4s, b1, b7) in Fig. 4D appear to be strange. In addition, regulatory subunits (Rpn1, Rpn10, Rpn11) were slightly downregulation in Rpn13-cKO on the basis of SDS-PAGE, but little or no band of the 26S proteasome were detected in native PAGE. Is there a free 19S-RP? I’m concerned that only the 26S proteasome bands (single-cap or double-cap), but not the 20S proteasome band, may be detected in the native-PAGE and blotting conditions (Fig. 4D). The 20S proteasome or 19S RP bands might pass through the PVDF membrane during blotting in case of low concentration of gel. If you will shorten the blotting time, you may be able to detect the free 19S RP and the 20S proteasome.
What are the intense bands of the 20S subunits (a4s, b1, b7) in Rpn13-cKO as shown in Fig. 4D? Is there any evidence that these intense bands are 26S proteasome or the 20S proteasome associated with other proteins?
Furthermore, the immunoblot data after SDS-PAGE using the antibodies against Rpn11, UCH37, USP14, a4S and PA200 must be necessary.
- Figure 4B: “19S-20S-19S”, “PA200-20S-19S”, and “PA200-20S”
Is there any convincing evidence that the above compositions of the 20S CP and 19S RP are correct? In order to know the composition, 2D-PAGE (First native-PAGE; second SDS-PAGE) is useful: please see the following reference (Sawada et al., FEBS letters 335, 207-212 (1993)). Immunoblotting after 2D-PAGE would be useful to identify the composition of each band in native PAGE. Such an experiment would also answer the previous questions. The second broad band may be the mixture of 19S-20S and PA200-20S, which can be distinguished by 2D-PAGE
3) In order to conclude that Rpn13 and Rpn10 play a non-redundant essential role in germ cell formation and fertility, the expression of Rpn13 in Rpn10-cKO mice should be examined. However, there is no information about the translational level of the 20S subunits and the regulatory subunits in Rpn10-cKO mice. The immunoblotting after SDS-PAGE (and native PAGE, if possible) must be necessary using the antibodies against Rpn1, Rpn10, Rpn11, Rpn13, UCH37, Usp14, P200, and 20S subunits (a4s, b1, b7), which is similar to Figure 4.
Minor points:
L109: “sum-LLVY-amc” should be amended to “succinyl-Leu-Leu-Val-Tyr-7-amido-4-methylcoumarin (Suc-LLVY-AMC)”.
Figure 1B: “10 0bp” --> “100 bp”
L230-231: "succinyl-LLVY-amino-4-methycoumain (AMC)” should be replaced with “Suc-LLVY-AMC”, since the abbreviation is spelled out in Line 109.
L245, L247, Figures 4B, 4C: “Suc-LLVY-amc” --> Suc-LLVY-AMC
L243: In native PAGE, please describe the percentage of native separating gel.
L254: “between exons 2 and 6” should be modified to “between exons 1 and 6”.
Figure 3F: There is no scale bar in the figure and the size description in the figure legend.
Figure 6C: There is no description of scale bar size in the legend. In addition, scale bar in the inset figures (right side) is not needed or indicate the size in the legend.
Figure 6G: Is the position of the scale bar OK? This means that the scale bar is in the inset figure. So, I suppose that the position of the scale bar is under the 3rd photos from the left.
Figure 6G (right) and S3 (right): The black (closed) column must be changed to while (open) column to unify the style, if there is no special reason. The explanation of each bar in the figures is not necessary.
Table S1: “Rpn13flox” and “Mouse” are bold letters. So, please change to regular letters.
Figure S1F and S1G: The scale bars in the inset figures (right side) are not needed. Alternatively, please indicate the size in the inset figures in the figure legend.
Author Response
Reviewer 2
Wan-Yu Yue et al. investigated the specific roles of Rpn10 and Rpn13, which are 19S regulatory subunits of the 26S proteasome, in primordial germ cells (PGCs) using the Cre-LoxP-mediated conditional gene knockout mice. They showed that the conditional deletion of either Rpn10 or Rpn13 in PGCs results in infertility in both male and female mice and that germ cells in testes and ovaries all decreased dramatically in the Rpn13-cKO mice. Specifically, deletion of Rpn13 in PGCs disrupted the assembly of the 26S proteasome, reduced the number of PGCs, and blocked meiosis of spermatocytes at zygotene stage during prophase I, whereas deletion of Rpn10 in PGCs sharply reduced PGC migration. They stated that their findings are important for understanding the roles of Rpn10 and Rpn13 in germ cell development and the related reproductive diseases.
Major points:
Although the previous paper (Ref. 10) reported that Rpn10 and Rpn13 play a redundant role in the degradation of ubiquitinated proteins in mouse liver, Yue et al. revealed that Rpn13 and Rpn10 showed a non-redundant essential role in germ cell formation and fertility. Furthermore, the phenotypes of cKO of Rpn13 and Rpn10 appear to be a little bit different, suggesting the different roles of Rpn13 and Rpn10 in PGC. These findings are interesting and valuable for the studies on the specific roles of proteasomal Rpn13 and Rpn10 in germ cell differentiation and sexual reproduction.
Before further consideration, the authors are requested to answer the following questions.
Responses: We greatly appreciate the reviewer for critical comments and suggestions, which are addressed point-to-point as below.
- L232-237 and Figure 4:
It is described that “Deletion of Rpn13 in PGCs dramatically reduced the levels of 19S subunits, including the ubiquitin receptors (Rpn13, Rpn1 and Rpn10) and deubiquitinating enzymes (UCH37, Usp14 and Rpn11), in addition to the proteasome activator PA200, and increased the levels of 20S subunits, including b1, b7 and the spermatoproteasome subunit a4s, as analyzed by immunoblotting following native-PAGE (Figure 4D)”.
However, the level of b7 in Fig. 4A (immunoblotting following SDS-PAGE) appears to be slightly downregulated in Rpn13-cKO, although it appears to be upregulated in native PAGE. Why? The authors should explain this apparent discrepancy. How many proteins were loaded in each lane in SDS-PAGE and native-PAGE?
Although the results of Fig. 4A seem to be reasonable, the results (a4s, b1, b7) in Fig. 4D appear to be strange. In addition, regulatory subunits (Rpn1, Rpn10, Rpn11) were slightly downregulation in Rpn13-cKO on the basis of SDS-PAGE, but little or no band of the 26S proteasome were detected in native PAGE. Is there a free 19S-RP? I’m concerned that only the 26S proteasome bands (single-cap or double-cap), but not the 20S proteasome band, may be detected in the native-PAGE and blotting conditions (Fig. 4D). The 20S proteasome or 19S RP bands might pass through the PVDF membrane during blotting in case of low concentration of gel. If you will shorten the blotting time, you may be able to detect the free 19S RP and the 20S proteasome.
What are the intense bands of the 20S subunits (a4s, b1, b7) in Rpn13-cKO as shown in Fig. 4D? Is there any evidence that these intense bands are 26S proteasome or the 20S proteasome associated with other proteins?
Responses: Native gel in Figure 4D shows an increased level of β7 subunit in the 26S proteasome complex in Rpn13-cKO testes, but SDS gel displays a decrease level of total β7, including the free form of β7 subunit in Figure 4A. There is probably a negative feedback regulation of the free β7 expression by the increase of β7 in the assembled 26S proteasome. Additional discussion on this point has now been included in the revised version (lines 265-267). 40 and 38 µg proteins were loaded in each lane in SDS-PAGE and native-PAGE, respectively.
In the revised Fig. 4B, we have now provided the band for PA200-20S proteasome in the native PAGE following activation of the 20S by 0.02% SDS (PMID: 7697125, PMID: 33262216).
Furthermore, the immunoblot data after SDS-PAGE using the antibodies against Rpn11, UCH37, USP14, a4S and PA200 must be necessary.
Responses: We have now added SDS-PAGE/Western blot for Rpn11, USP14 and UCH37 (revised Fig. 4A), all of which are downregulated.
- Figure 4B: “19S-20S-19S”, “PA200-20S-19S”, and “PA200-20S”
Is there any convincing evidence that the above compositions of the 20S CP and 19S RP are correct? In order to know the composition, 2D-PAGE (First native-PAGE; second SDS-PAGE) is useful: please see the following reference (Sawada et al., FEBS letters 335, 207-212 (1993)). Immunoblotting after 2D-PAGE would be useful to identify the composition of each band in native PAGE. Such an experiment would also answer the previous questions. The second broad band may be the mixture of 19S-20S and PA200-20S, which can be distinguished by 2D-PAGE
Responses: The locations of proteasomal 26S and 20S subunits are correspond to the activity sites of the 26S and the 20S, the latter of which can be activated by 0.02% SDS (PMID: 7697125). In the revised Fig. 4B, we have now provided the band for PA200-20S proteasome in the native PAGE following activation of the 20S by 0.02% SDS, judged from our previous study (PMID: 33262216).
3) In order to conclude that Rpn13 and Rpn10 play a non-redundant essential role in germ cell formation and fertility, the expression of Rpn13 in Rpn10-cKO mice should be examined. However, there is no information about the translational level of the 20S subunits and the regulatory subunits in Rpn10-cKO mice. The immunoblotting after SDS-PAGE (and native PAGE, if possible) must be necessary using the antibodies against Rpn1, Rpn10, Rpn11, Rpn13, UCH37, Usp14, P200, and 20S subunits (a4s, b1, b7), which is similar to Figure 4.
Responses: Unlike Rpn13-cKO mice, Rpn10-cKO mice cannot develop germ cells in testes. Thus, we could not analyze proteasome subunits as in Figure 4.
Minor points:
L109: “sum-LLVY-amc” should be amended to “succinyl-Leu-Leu-Val-Tyr-7-amido-4-methylcoumarin (Suc-LLVY-AMC)”.
Figure 1B: “10 0bp” --> “100 bp”
L230-231: "succinyl-LLVY-amino-4-methycoumain (AMC)” should be replaced with “Suc-LLVY-AMC”, since the abbreviation is spelled out in Line 109.
L245, L247, Figures 4B, 4C: “Suc-LLVY-amc” --> Suc-LLVY-AMC
L243: In native PAGE, please describe the percentage of native separating gel.
L254: “between exons 2 and 6” should be modified to “between exons 1 and 6”.
Figure 3F: There is no scale bar in the figure and the size description in the figure legend.
Figure 6C: There is no description of scale bar size in the legend. In addition, scale bar in the inset figures (right side) is not needed or indicate the size in the legend.
Figure 6G: Is the position of the scale bar OK? This means that the scale bar is in the inset figure. So, I suppose that the position of the scale bar is under the 3rd photos from the left.
Figure 6G (right) and S3 (right): The black (closed) column must be changed to while (open) column to unify the style, if there is no special reason. The explanation of each bar in the figures is not necessary.
Table S1: “Rpn13flox” and “Mouse” are bold letters. So, please change to regular letters.
Figure S1F and S1G: The scale bars in the inset figures (right side) are not needed. Alternatively, please indicate the size in the inset figures in the figure legend.
Responses: Thanks a lot for these valuable suggestions, and we have now corrected all of the above issues.
Reviewer 3 Report
Comments and Suggestions for Authors
This is an interesting study and appears to be carefully conducted.
There are some points that need attention:
line 126: To explore the underlying mechanism.. please specify which mechanism(s), as the sentence is not clear.
lines 167-169: in my view in Fig. 3A, although I may agree that the number of spermatogonia does not change, the distribution within the tubule appears different or the magnification in the upper panel is different. Something similar appears to happen for SYCP3 in Fig. 3B.
In Fig. 3F, tunel positivity is hardly seen in both upper and lower panels. Maybe the Authors may chose a different image.
In Fig. 4D the mol. weight should be indicated
In Fig. 6, H&E staining is shown whereas for Fig. 2 it is shown in a supplementary figure: why? I suggest to include H&E also in Fig. 2.
The discussion contains some parts that belong more to the introduction than to the discussion. The Authors should discuss their results and their novelties, more than reporting the roles played by Rpn proteins (beginning of the discussion, lines 231-238: this part can be in the introduction rather or can be shortened).
in the line 241: This study shows.. maybe better Our study shows.
Author Response
Reviewer 3
This is an interesting study and appears to be carefully conducted.
There are some points that need attention:
Responses: We greatly appreciate the reviewer for critical comments and suggestions, which are addressed point-to-point as below.
line 126: To explore the underlying mechanism. please specify which mechanism(s), as the sentence is not clear.
Responses: Thanks for the advice. We have now changed it into “To explore the mechanism underlying the role of Rpn13 in reproduction,”
lines 167-169: in my view in Fig. 3A, although I may agree that the number of spermatogonia does not change, the distribution within the tubule appears different or the magnification in the upper panel is different. Something similar appears to happen for SYCP3 in Fig. 3B.
Responses: Since sizes of the seminiferous tubules in control and Rpn13-cKO are different, the pictures look different, but they’re actually the same magnification.
In Fig. 3F, tunel positivity is hardly seen in both upper and lower panels. Maybe the Authors may chose a different image.
Responses: We have replaced the original one.
In Fig. 4D the mol. weight should be indicated
Responses: Fig. 4D indicates the proteasomal subunits in the 26S or 20S proteasome complex, which do not correspond to the actual molecular weight in native PAGE. The locations of proteasomal 26S and 20S subunits are correspond to the activity sites of the 26S and the 20S, the latter of which can be activated by 0.02% SDS (PMID: 7697125). In the revised Fig. 4B, we have now provided the band for PA200-20S proteasome in the native PAGE following activation of the 20S by 0.02% SDS, judged from our previous study (PMID: 33262216).
In Fig. 6, H&E staining is shown whereas for Fig. 2 it is shown in a supplementary figure: why? I suggest to include H&E also in Fig. 2.
Responses: Thanks for the suggestion. H&E has now been moved from Supplementary Figures into the revised Fig. 2.
The discussion contains some parts that belong more to the introduction than to the discussion. The Authors should discuss their results and their novelties, more than reporting the roles played by Rpn proteins (beginning of the discussion, lines 231-238: this part can be in the introduction rather or can be shortened).
Responses: Thanks for the suggestions, and related changes have been made now (line 51-59).
in the line 241: This study shows. maybe better Our study shows.
Responses: We have replaced “This study shows” into “Our study shows”.
Reviewer 4 Report
Comments and Suggestions for Authors
See attached PDF file.

Author Response
Reviewer 4
Yue et al. investigate the effect RPN13 or RPN10 deficiency in primordial germ cells in mice. Deletion of either Rpn10 or Rpn13 in PGCs results in infertility in mice. Deletion of Rpn13 in PGCs blocks meiosis of spermatocytes at zygotene stage during prophase I, whereas deletion of Rpn10 in PGCs sharply reduces PGC migration.
Responses: We greatly appreciate the reviewer for critical comments and suggestions, which are addressed point-to-point as below.
Abstract: Define cKO (line 19)
Responses: Thanks for suggestion, we have now defined the cKO in Abstract.
Introduction:
-starting from line 32: Indicate that embryonic stages of mice are described.
Responses: “in mice” have now been added.
- line 44: PA28α/β, PA28γ and PA200 are regulators of the proteasome and not a categorization of proteasomes.
Responses: We correct it as “…… with four types of activators, namely 19S regulatory particle, PA28αβ, PA28γ and PA200.
Material and methods:
- line 94: What is NAPP
Responses: Na4P2O7, and we have now corrected it.
- line 109: Suc-LLVY-AMC
- line 115: are
Responses: Thanks, and we have now corrected them.
Results:
- Fig. 1A: Show the whole mutated allele; similar to Figure 5. Indicate where primers used for Fig. 1B bind.
Responses: We have now added the vertical line to indicate the position for Rpn10 or Rpn13 flox detection.
- Fig. 1B: 100 bp not 10 0bp
Responses: Thanks a lot, and we have replaced it to 100 bp.
- Fig. 1C: Indicate how many embryos were used for quantification. Were several sections of the same embryo used for quantification?
Responses: As added in the revised version, there were 5 and 4 embryos in control and cKO groups at E8.5, respectively. There were 2 embryos at E9.5, 4 embryos at E10.5, and 2 embryos at E11.5 in both control and cKO groups (line 150-152 in the revised version).
Figure 4 is not that convincing to me. Some controls are missing:
Fig.4: How many times were western blots repeated? How many samples were used for quantification?
Fig. 4B: Arrows: How do you know this is 19S-20S-19S or PA200-20S-19S or PA200-20S?
Responses: The locations of proteasomal 26S and 20S subunits are correspond to the activity sites of the 26S and the 20S, the latter of which can be activated by 0.02% SDS (PMID: 7697125). In the revised Fig. 4B, we have now provided the band for PA200-20S proteasome in the native PAGE following activation of the 20S by 0.02% SDS, judged from our previous study (PMID: 33262216).
There were 2 or 3 pairs of control and cKO mice for at least two repeated western blot repeats and 4 or 6 samples used for quantification.
Fig. 4D indicates that PA200 is present in both bands. Please also show the 20S. Is the 20S increased in the rpn13-cKO? Why is the PA200-20S band less intensive in the rpn13-cKO? A loading control should be shown.
Responses: We had provided the results for the 20S subunits, β1 and β7, both of which showed an increase in the Rpn13-cKO (Fig. 4D). It remains unknown why Rpn13 deletion in PGC leads to a reduction in the PA200-20S band. Unlike SDS-PAGE, a loading control is hard to be shown in native PAGE.
Fig. 4D: Why is alpha4s, beta1, and beta7 at the size of PA200-20S? Why is rpn1 so low? A loading control should be used.
Responses: Because this is a native gel, subunits in the proteasome complex PA200-20S would not separate, and these 20S subunits would appear at the size of the PA200-20S.
We are sorry for failing to introduce this difference for Rpn1 in the previous version. Actually, they were from two different gels. The bands for Rpn1 were the results in which the gel run a little longer than the gel for other subunits in Fig. 4D. In the revised version, we have now described them.
Fig. 5: Indicate the number of samples used for quantification.
Responses: As added in the revised version, there were 3 and 4 embryos in control and cKO female groups, respectively, at E13.5. There were 2 embryos at E10.5 and E11.5, and 3 embryos at E13.5 for male in both control and cKO groups (lines 299-301).
Fig. 5A: Why is there a loxP site between 9 and 10 after excision?
Responses: This was an error, and we have now deleted it.
Supplementary files: Indicate number of experiments and number of samples used for quantification.
S1A: The expression “total litter size§ is not clear. Do you mean total number of litters?
Responses: Number of experiments and number of samples used for quantification have now been included in the revised version.
Yes, litter numbers. A correction has now been made.
Round 2
Reviewer 2 Report
Comments and Suggestions for Authors
The following point has not been amended. Please correct this point.
Figure 4C: suc-LLVY-amc --> Suc-LLVY-AMC